

# Dynamics of nitrogen and active nitrogen components across seasons under varying stand densities in a *Larix principis-rupprechtii* (*Pinaceae*) plantation

Junyong Ma, Hairong Han, Wenwen Zhang and Xiaoqin Cheng

Beijing Forestry University, Key Laboratory of Ministry of Forest Cultivation and Conservation of Ministry of Education, Beijing, China

## ABSTRACT

Changes in the concentration of soil nitrogen (N) or its components may directly affect ecosystem functioning in forestry. Thinning of forest stands, a widely used forestry management practice, may transform soil nutrients directly by altering the soil environment, or indirectly by changing above- or belowground plant biomass. The study objectives were to determine how tree stem density affects the soil N pool and what mechanisms drive any potential changes. In this study, N and its active components were measured in the soil of a *Larix principis-rupprechtii* plantation across two full growing seasons, in 12 (25 × 25 m) plots: (low thinning, removal of 15% of the trees, three plot repetitions), moderate thinning (MT) (35% removal) and heavy thinning (HT) (50% removal) and no thinning control. Environmental indices, like the light condition, soil respiration, soil temperatures, and prescription, were measured in the plots also. Results indicated that soil total nitrogen (STN) was affected by tree stem density adjustments in the short-term; STN generally increased with decreasing tree stem density, reaching its highest concentration in the MT treatment before decreasing in HT. This pattern was echoed by the DON/STN ratio dissolved organic nitrogen (DON) under MT. A lower DON/STN was measured across the seasons. Microbial biomass nitrogen (MBN) and the SOC/STN (soil organic carbon (SOC)) ratio and density treatments influenced MBN concentration and inhibited SOC/STN. MT tended to accumulate more STN, produce lower DON/STN and had a generally higher microbial activity, which may be partly ascribed to the higher MBN value, MBN/STN ratio and lower DON/STN. The water conditions (soil moisture), light and soil temperatures could partly be responsible for the N pool dynamic in the different density treatments.

## INTRODUCTION

Forest ecosystems have often been proposed to play a part in the effective mitigation of climate change (*Canadell & Raupach, 2008*; *Miles & Kapos, 2008*). Playing a major role in global nutrient cycles and providing regulating and supporting services, soils are one of the most important components of forest ecosystems (*Bravo-Oviedo et al., 2015*).

Corresponding author
Hairong Han,
hanhr6015@bjfu.edu.cn

Previous studies have suggested that increasing levels of nitrogen (N) deposition could impact the sustainability of carbon (C) sinks in forest ecosystems (*Townsend et al., 1996*) because of interactions between the carbon and nitrogen cycles (*Rastetter, Ågren & Shaver, 1997*). However, due to the complexity of the interactions between both cycles, how these cycles are coupled remains poorly understood (*Mcguire, Melillo & Joyce, 1995*). Studies (*Aerts, Bodegom & Cornelissen, 2012*; *Wieder et al., 2013*) have shown that soil total nitrogen (STN), which has been widely studied in forest ecosystems (*Hafner & Groffman, 2005*; *Guan et al., 2015*), and other land use conditions (*Lehrsch, Sojka & Koehn, 2012*; *Zhao et al., 2017*; *Wang, Zhuang & Zhu, 2017*), respond to soil organic matter input. Therefore, aboveground changes may potentially alter N pools in temperate forests.

Thinning treatments are frequently utilized in forest management to promote undergrowth renewal, increase biodiversity and improve soil fertility (*Pariona, Fredericksen & Licona, 2003*). Management of stem density has been shown to be important for maintaining forest ecosystem services and long-term productivity, and is thus a focus of much scientific study (*Jackson, Fredericksen & Malcolm, 2002*; *Crow, Buckley & Nauertz, 2002*).

More than 80% of the N in soil exists in organic form (*Schulten & Schnitzer, 1997*). However, recent studies in terrestrial ecosystems have mainly focused on inorganic forms, such as ammonium ($NH_4^+$) and nitrate ($NO_3^-$) (*Sigua & Coleman, 2006*). STN is strongly correlated with the amount of available N in soil, and thus can influence soil microbial activity and humus formation (*Bravo-Oviedo et al., 2015*). Dissolved organic nitrogen (DON) availability may dictate the structure and presence of bacterial communities (*Ren et al., 2016*), change rapidly in response to environmental factors and affect soil nutrient cycling (*Iqbal et al., 2010*). Although total soil microbial biomass nitrogen (MBN) tends to be low in absolute value, its turnover represents a significant contribution to the global nitrogen cycle (*Jenkinson, 1988*). The MBN reflects the activity of microorganisms (*Wardle, 1992*; *Jiang et al., 2010*). Global stocks of soil organic carbon (SOC) recently reached 2,344 Pg (*Stockmann et al., 2013*), a large percentage of the global soil carbon pool, concentrated in forest soils (*Houghton, 1995*; *Dixon et al., 1994*). Due to the close relationship between C and N in forest soils (*Tateno & Chapin, 1997*; *Cleveland & Liptzin, 2007*), the SOC/STN ratio acts as an index of the degree of correlation between C and N availability (*Ge et al., 2013*), as well as a sensitive indicator of soil quality (*Gravel et al., 2010*). This SOC/STN ratio can also detect plant growth (*Zhang et al., 2011*; *Wieder et al., 2013*).

Tree stem density adjustment via thinning is a common management practice in forest plantations; this widely used approach can affect the growth of the forest stand (*Duan et al., 2010*), aboveground plant biomass (*Archer, Miller & Tanner, 2007*) and understory biological diversity (*Karlsson et al., 2002*; *Lähde et al., 2002*). Thinning regulates the distribution of open growing space so that standing trees may benefit from reduced competition, increased growth and tree health (*Smith et al., 1997*; *Jandl et al., 2007*). Afforestation increases soil nitrogen accumulation and modifies nitrogen availability for micro-organismal growth (*Deng et al., 2014*), thereby potentially influencing elemental cycles in terrestrial ecosystems (*Li, Niu & Luo, 2012*; *Li et al., 2014*). Studies (*Aerts, Bodegom & Cornelissen, 2012*; *Wieder et al., 2013*) have also shown that

soil N responds to changes in soil organic matter inputs, which can then impact microbial processes. While many studies have focused on the soil carbon cycle in forest ecosystems (*Lal, 2004*; *Zou et al., 2005*; *Ares, Neill & Puettmann, 2010*), rather less attention has been paid to the relationship between C and N. Knowledge of how the active organic form of soil nitrogen varies with stand tree stem density and how SOC and STN are mechanistically linked is lacking.

In this study, within-growing-season variation in soil active nitrogen components was quantified for four different stand densities within a *Larix principis-rupprechtii* plantation located in a Northern Chinese montane secondary forest. Study hypotheses were first that adjustments in the tree stem density would affect STN, and second that soil N-components would play an important role in N cycling. The specific objectives were to determine: (1) how STN varies with stand tree stem density; (2) the contributions of each soil nitrogen component to variation in the nitrogen pool overall under different stand densities and in different seasons; and (3) how the environmental factors change according to thinning treatments and how those differences may influence the N pools.

## MATERIALS AND METHODS

### Study area and experimental design

The study was carried out in a plantation on Mount Taiyue in Shanxi, North China (112°00′ 47″E, 36°47′05″N; 112°01′–112°15′E, 36°31′–36°43′N; elevation 2,273–2,359 m above sea level). This artificial forest is dominated by *L. principis-rupprechtii* and has been protected since it was planted in the 1980s. The climate is the continental monsoon type with a humid, rainy summer and a cold, snowy winter. Mean annual air temperature is 8.7 °C, with an average minimum temperature of −10.4 °C in January and an average maximum of 17.4 °C in July. The frost-free period lasts an average of 125 days, with the earliest frost generally in October and latest frost in April. Average annual rainfall ranges between 600 and 650 mm·year$^{-1}$, with precipitation occurring mainly from July to September. The soil type in the study plantation is Haplic luvisol, ranging from 50 to 110 cm thick, according to the world reference base soil classification system (*IUSS Working Group WRB, 2006*).

Sampling was performed in stands selected to reflect average altitude (2,316 m), grade (24.3 ± 0.83°), slope direction (North) and soil conditions within the plantation. Measurements of these characteristics did not vary significantly among selected stands before the experimental treatments were applied (Table 1). After quantifying the initial characteristics of each quadrat, 12 (25 × 25 m) quadrats, or "sample areas," were designated for four treatments (one, no thinning control (CK) contrast contained) in July 2010. Three control quadrats had initial stand densities averaging 2,160 ± 12 stems ha$^{-1}$ and no thinning treatment was applied (CK). Three quadrats each were selected to undergo the following treatments: 15% thinning (low thinning (LT)) with tree stem density adjusted to 1,834 ± 12 stems ha$^{-1}$ (mean of three replications); 35% thinning (moderate thinning (MT)) with tree stem density adjusted to 1,418 ± 7 stems ha$^{-1}$; and 50% thinning (heavy thinning (HT)) with tree stem density adjusted to 1,089 ± 3 stems ha$^{-1}$. The trees that were cut for thinning were removed from the plots, but the understory shrub and herb layers remained untouched. The dominant overstory vegetation in all

**Table 1 Average characteristic measurements of experimental stands for density adjustment treatments in a 35-year-old *Larix principis-rupprechtii* plantation.**

| Treatment | Stems (ha$^{-1}$) | Thinning (%) | Slope gradient (°) | Mean height (m) | Mean DBH (cm) | Total soil phosphorus (g kg$^{-1}$) | Soil bulk density g cm$^{-3}$ | Mechanical composition (%) | | |
|---|---|---|---|---|---|---|---|---|---|---|
| | | | | | | | | <0.002 mm | 0.002–0.05 mm | 0.05–2.00 mm |
| CK | 2,173 (±12) | 0 | 25 (±3.6) | 14.5 (±1.21) | 13.3 (±1.29) | 0.50 (±0.032) | 0.91 (±0.070) | 20.83 (±4.263) | 30.40 (±0.589) | 48.77 (±4.246) |
| LT | 1,834 (±12) | 15 | 25 (±3.6) | 19.3 (±1.21) | 14.9 (±1.29) | 0.51 (±0.032) | 0.87 (±0.070) | 22.07 (±4.263) | 29.90 (±0.589) | 48.03 (±4.246) |
| MT | 1,418 (±7) | 30 | 23 (±0.5) | 16.6 (±0.21) | 16.3 (±0.02) | 0.60 (±0.071) | 0.95 (±0.024) | 18.27 (±2.117) | 31.93 (±2.738) | 49.80 (±1.768) |
| HT | 1,089 (±3) | 50 | 24 (±2.0) | 16.9 (±0.31) | 17 (±0.65) | 0.57 (±0.034) | 0.86 (±0.010) | 17.43 (±1.156) | 33.60 (±2.570) | 48.97 (±3.703) |

**Notes:**
Standard errors of the mean are presented within parenthesis. Density adjustments and characteristics were measured in July 2012. Total phosphorus, bulk density, mechanical composition and bulk density of soil was measured in July 2015 (means ± SD, $n$ = 3).
Treatments: CK, no thinning control; LT, low thinning; MT, moderate thinning; HT, heavy thinning.

**Table 2 The environmental factors of *L. principis-rupprechtii* plantation with different thinning treatments.**

| Environmental factors | Year | CK | LT | MT | HT |
|---|---|---|---|---|---|
| Soil respiration (gC m$^{-2}$) | 2015 | 297.6 ± 22.1[a] | 280.43 ± 31.97[a] | 391.1 ± 40.6[a] | 356.7 ± 33.6[a] |
| | 2016 | 421.6 ± 47.3[a] | 391.08 ± 70.42[a] | 507.5 ± 55.4[a] | 438.8 ± 45.3[a] |
| PPFD total under (MJ·m$^{-2}$·d$^{-1}$) | 2015 | 4.6 ± 0.5[c] | 5.9 ± 0.47[b] | 6.5 ± 0.5[b] | 9.7 ± 0.5[a] |
| | 2016 | 4.8 ± 0.3[c] | 5.86 ± 0.21[b] | 6.4 ± 1.0[b] | 8.1 ± 0.4[a] |
| PPFD total over (MJ·m$^{-2}$·d$^{-1}$) | 2015 | 27.9 ± 1.2 | 28.3 ± 1.23 | 28.4 ± 0.3 | 28.5 ± 0.6 |
| | 2016 | 28.87 ± 1.07[a] | 29.25 ± 0.14[a] | 29.5 ± 1.1[a] | 30.4 ± 1.2[a] |
| Soil temperature (°C) | 2015 | 6.1 ±0.1[b] | 6.3 ±0.3[b] | 7.0 ± 0.2[a] | 6.5 ± 0.2[ab] |
| | 2016 | 7.7 ± 0.4[a] | 7.7 ±1.2[a] | 8.5 ± 0.8[a] | 7.8 ± 0.6[a] |
| Soil moisture (%) | 2015 | 22.1 ± 0.8[b] | 24.2 ±3.3[a,b] | 27.1 ± 1.9[a,b] | 28.7 ± 2.1[a] |
| | 2016 | 22.7 ± 1.4[b] | 24.1 ± 2.9[a,b] | 25.0 ± 2.2[a,b] | 28.2 ± 1.2[a] |

**Note:**
Soil respiration: carbon flux of soil respiration; PPFD total over: photosynthetic photon flux density over the forest; PPFD total under: photosynthetic photon flux density under the forest. The soil respiration, soil temperature, soil moisture was measured in the vegetation growing seasons and the values were the means of 7 months from April to October. PPFD was measured in the summer seasons, July each year. Different superscripts indicate significant difference at $p < 0.05$ in thinning treatments, $n$ = 3.

stands was 35-years-old *L. principis-rupprechtii*. Shrub species included *Elaeagnus umbellata* and *Rubus parvifolius*. Herbaceous species included *Carex rigescens* and *Dendranthema chanetii*. Detailed treatment characteristics are presented in Tables 1 and 2.

## Sampling and chemical analysis

Total soil carbon and nitrogen concentration were determined from soil samples collected from treatments at 0–10, 10–20, 20–30, 30–40, and 40–50 cm depths using a cylindrical soil auger. Samples were collected at three times during the 2015 growing season: spring, summer, and autumn. Snow cover and freezing prevented collection of soil samples in the winter. Soil samples were collected from nine randomly chosen locations within

each quadrat and then combined according to depth to form one homogenous composite sample per depth. Visible stones and organic residues were removed and each sample was sieved through a two mm mesh prior to chemical analyses. After sifting, each composite soil sample was divided into two subsamples. One subsample was stored in a 4 °C incubator until DON and MBN concentration could be determined. The second sample was air-dried and passed through a 0.25 mm sieve before determining SOC concentration and STN concentration. Then it was passed through a two mm sieve for soil pH.

Soil organic carbon and STN concentrations were determined by dry combustion using an elemental analyzer (FLASH 2000 CHNS/O; Thermo Fisher Scientific, Waltham, MA, USA). The MBN concentration was measured using an $HCl_4$-fumigation extraction technique; $10.0 \pm 0.5$ g of fresh soil was fumigated with $HCl_4$, then extracted with 40 mL of $0.5 \text{ mol} \cdot L^{-1}$ $K_2SO_4$, shaken for one h at $350 \text{ r min}^{-1}$, and filtered through a 0.45 μm membrane after centrifuging 5 min at $3,000 \text{ r min}^{-1}$. The filtrate concentration was quantified using a total organic carbon analyzer (Multi N/C 3000, Analytik; Jena, AG, Jena, Germany). The DON concentration was measured as the carbon concentration of non-fumigated soil samples (*Boyer & Groffman, 1996*).

MBN was calculated as:

$$MBN = E_C / k_{EC} \tag{1}$$

In (1) $E_C =$ (organic N extracted from fumigated soils) $-$ (organic N extracted from non-fumigated soils) and $k_{EC} = 0.54$.

The soil texture was analyzed using the pipette method (*Gee & Bauder, 1986*). Air-dried soil samples that had been passed through a one mm sieve were used for soil pH determination. Using a pH meter (Sartorius PB-10), pH was determined for a 1:2.5 soil-water mixture. Gravimetric soil water concentration was measured as mass lost after drying for 24 h at 105 °C. Meteorological data was collected from a small fixed weather station beside the sample area.

## Environmental factors in the density adjustment plots

Soil respiration was measured with an LI-8100 soil $CO_2$ flux system (LI-COR Inc., Lincoln, NE., USA) in the middle and at the end of each month during the three sampling seasons. Measurements were taken on 12 PVC collars in each plot during 10:00–17:00 h over a one-day period. The PVC collars in each plot were systematically arranged. Soil temperature and volumetric soil water content at five cm depth were concurrently measured near each PVC collar. Each PVC collar is 10 cm in diameter and five cm in height, inserted three cm below the surface of the soil.

Soil respiration was calculated as the average values measured twice each month during the vegetation growing season and the computation formula is as follows:

$$C_{RS} = \frac{R_S \cdot t \cdot C_{mol}}{10^6} \tag{2}$$

$C_{RS}$ (total carbon emission from soil respiration) $gC \cdot m^{-2}$; $R_S$ (Soil respiration), $\mu mol \cdot m^{-2} \cdot s^{-1}$; $t$ (time), s; $C_{mol}$, $12 \text{ g} \cdot mol^{-1}$. $C_{RS}$ in winter in this study accounted for 10% of the total carbon emissions from annual soil respiration (*Wang, Bond & Gower, 2002*).

The forest light environment measurements were collected in July 2015 and 2016. The canopy analyzer (WINSCANOPY 2010a; Instruments Regent Inc, Quebec, Canada) was used to measure total photosynthetic photon flux density (PPFD) over: PPFD over the forest and total PPFD under: PPFD under the forest. The plot was divided into three areas, left, middle, and right, each of which was divided into three sub areas, for a total of nine areas. The canopy analyzer was set up in the center of each sub area. Optical information was collected and the instrument software was used to analyze stand light environment (PPFD) back in the laboratory.

## Statistical analysis

SPSS 20.0 (IBM, Chicago, IL, USA) and R were used for statistical analyses (*R Core Team, 2018*). All data in the tables and figures are presented as means ($n = 15$, 3 plot repeats × 5 soil depths). A mixed-effects model was used to consider the relationships between year, season and treatment. To improve the representativeness of the samples, we used the mean value of each soil property, averaging the values from each of the five layers, to represent the content of each value in the plot, based on the post hoc Tukey-HSD test using SPSS. One-way analysis of variance was used to examine the impact of thinning treatment in a certain season by the *T*-test, using SPSS. The least significant difference test was used to compare treatment means, with significant effects having $p < 0.05$. To examine the relationship between soil chemical variables, the data was pooled from the depths of 10–20, 20–30, 30–40, and 40–50 cm, the six sampling seasons and the 12 independent plots ($n = 360$), and was examined for Pearson relationships using R and the "Performance Analytics" package for the data visualization.

# RESULTS

## General environmental characteristics

The meteorological data recorded by an automatic meteorological station indicated that precipitation was significantly higher in the summer than in the spring or autumn. Both the air temperature and the 0–10 cm soil temperature were also highest in summer. In 2016, precipitation was 1.3 times greater than in 2015 (Fig. 1).

No significant difference was found in the total phosphorus, bulk density, or mechanical composition of soil among the different thinning treatments (Table 1). With greater tree stem density reduction, or thinning, the forest understory became much brighter from both direct and scattered light as well as total radiation. The total (PPFD under) in the understory increased in the LT, MT, and HT treatments, with respect to the control ($p < 0.05$; Table 2). Soil respiration was higher in the MT plots, but not significantly. The higher soil temperature measured in MT only differed from the other treatments significantly in 2015. Soil moisture was only significantly higher than CK in HT (Table 2).

The total biomass gradually decreased with increased thinning. The variance analysis showed that this difference was only significant between the CK and HT treatments ($p < 0.05$) (Table 3).

The understory species composition was relatively simple in this *L. principis-rupprechtii* plantation, with the understory vegetation in the CK containing nine families, 13 genera and 14 herbaceous species. Dominant plants included species of the *Compositae*,

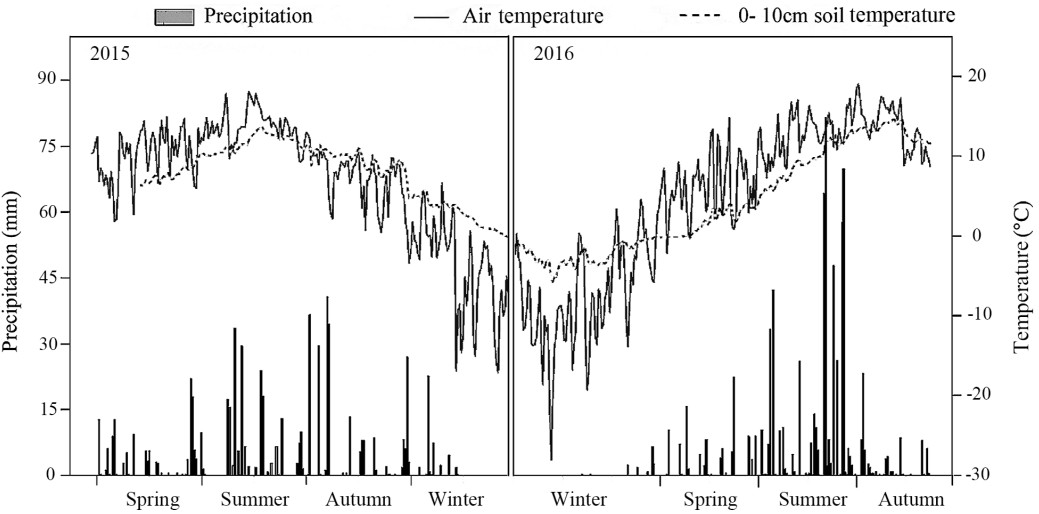

**Figure 1** Air temperature, 10 cm soil temperature and average precipitation in the study treatments in 2015 and 2016.                                                            

**Table 3 Biomass (t ha⁻¹) of the *L. principis-rupprechtii* plantation with different thinning treatments.**

| Components | Treatments | | | | |
|---|---|---|---|---|---|
| | CK | LT | MT | HT | Mean |
| Tree layer | 189.58 ± 2.06[a] | 159.17 ± 7.59[b] | 144.98 ± 5.58[b,c] | 135.55 ± 3.44[c] | 157.32 ± 23.60 |
| Understory layer | 2.24 ± 0.25[a] | 2.83 ± 0.42[a] | 5.56 ± 1.14[a] | 6.95 ± 1.57[a] | 4.40 ± 2.23 |
| Litter layer | 61.88 ± 10.53[a] | 57.71 ± 14.55[a] | 62.35 ± 14.49[a] | 60.84 ± 19.38[a] | 60.70 ± 2.09 |
| Total | 253.70 ± 8.72[a] | 219.70 ± 22.48[a,b] | 212.90 ± 17.33[a,b] | 203.33 ± 18.67[b] | 222.41 ± 21.92 |

**Note:**
Values mean ± SD; different superscripts indicate a significant difference at $p < 0.05$ in thinning treatments; Biomass data was collected in July 2014 and the total biomass is the sum of the three layers (tree, understory and little layer).

*Ranunculaceae*, and *Rosaceae* families. In the plots undergoing thinning treatments, understory plant species richness increased with decreasing tree stem density. Overall, the highest species richness was recorded in the MT treatment (Table S2). Soil nutrients decreased significantly with soil depth and generally accumulated to higher levels as the summer progressed (Table S2).

## Soil total nitrogen

Tree stem density effects on STN were significant in five sampling seasons out of six, (Fig. 2A). In both 2015 and 2016, a significantly higher summer STN was measured in the MT than the LT and the CK. STN was analyzed separately in each season. In spring 2015 ($p = 0.0027$), (g N Kg⁻¹): MT (3.1 ± 0.21) > HT (2.9 ± 0.33) > CK (2.5 ± 0.05) > LT (2.3 ± 0.11 g). In summer 2015 ($p = 0.002$), (g N Kg⁻¹): MT (3.6 ± 0.04) > HT (3.4 ± 0.21) > LT (2.9 ± 0.08) > CK (2.7 ±0.19). In autumn 2015 ($p = 0.110$), (g N Kg⁻¹): HT (3.2 ± 0.42) > MT (3.1 ± 1.197) > CK (2.7 ± 0.29) > LT (2.5 ± 0.97). Thus, STN was highest in spring and summer in the MT treatment, compared to other treatments. Mean STN concentrations were 25% higher in the MT (30% thinning) and HT (50% thinning) treatments than in the less severe thinning treatments (i.e., LT—15% thinning, and CK—0% thinning).

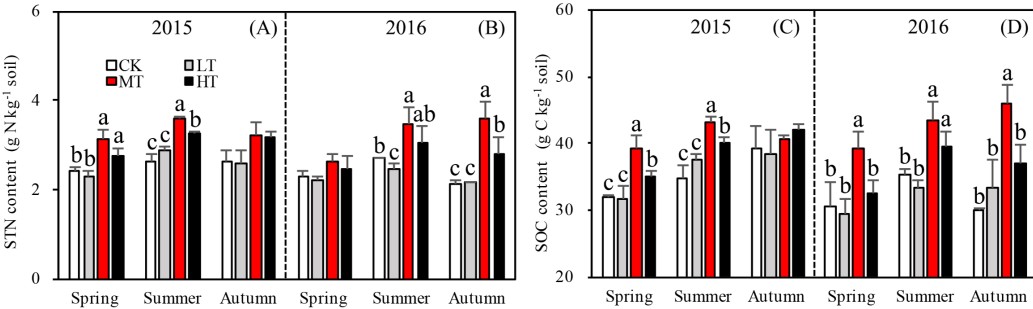

**Figure 2 Variation in the STN (A, B) and SOC (C, D) in different thinning treatments across the growing seasons in 2015 and 2016.** CK, the no thinning control treatments. LT, the low thinning treatments (15% thinning). MT, the moderate thinning sample treatments (35% thinning). HT, the heavy thinning sample treatments (50% thinning). STN, soil total nitrogen; SOC, soil total organic carbon. Each bar represents an average value across three replicate samples ($n = 15$), that is, three plots repeats × five soil depths. Error bars represent standard errors around the three plot repeats. Different lowercase letters demarcate a significant difference among different density adjustments within the same sampling season ($p < 0.05$). The same for Figs. 3–5.               

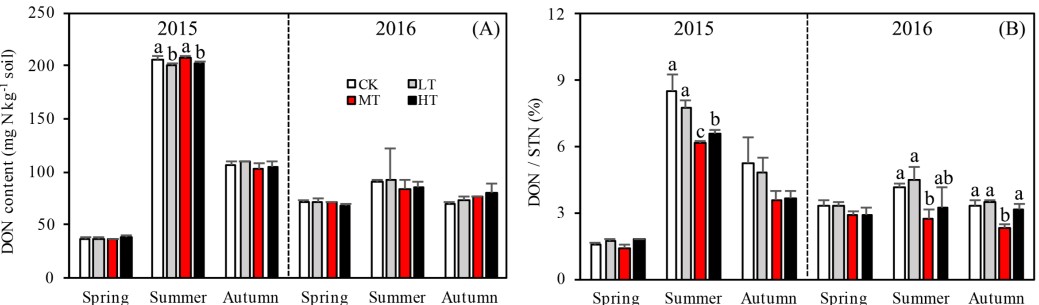

**Figure 3 Variation in the DON (A) and DON/STN (B) in different thinning treatments across the growing seasons.** CK, the no thinning control treatments. LT, the low thinning treatments (15% thinning). MT, the moderate thinning sample treatments (35% thinning). HT, the heavy thinning sample treatments (50% thinning). DON, dissolved organic nitrogen; STN, soil total nitrogen. Each bar represents an average value across three replicate samples ($n = 15$), that is, three plots repeats × five soil depths. Error bars represent standard errors around the three plot repeats. Different lowercase letters demarcate a significant difference among different density adjustments within the same sampling season ($p < 0.05$).               

In 2016, the response of STN content to density adjustment was similar to 2015, but there were bigger differences between the HT and MT treatments and the LT and CK treatments. In 2016, the tree stem density effects on STN were significant in spring ($p = 0.003$), summer ($p = 0.026$), and autumn ($p = 0.003$). Across the three sampling seasons, (g N Kg$^{-1}$): MT ($3.2 \pm 0.44$) > HT ($2.8 \pm 0.23$) > CK ($2.4 \pm 0.24$) > LT ($2.3 \pm 0.13$).

Accumulation of STN content was greater for the treatments with more thinning (MT, HT) than the treatments with less thinning (CK, LT) in both sampling years, resulting in 26.1%, 24.9%, and 22.5% increases between less thinned and more thinned treatments in spring, summer, and autumn, respectively, in 2015 (Fig. 3A) and resulting in 12.5%, 26.3%, and 48.9% increases between less thinned and more thinned treatments in spring, summer, and autumn, respectively, in 2016 (Fig. 3B).

**Table 4 Mixed-effects model analysis of soil properties to year, season, and density (or thinning treatment).**

| Factors | Df | STN | SOC | SOC/STN | DON | MBN | DOC | MBC | pH | MBNSTN | DON/STN |
|---|---|---|---|---|---|---|---|---|---|---|---|
| | | $p$ | $p$ | $p$ | $p$ | $p$ | $p$ | $p$ | $p$ | $p$ | $p$ |
| Treatment | 3 | <0.001 | <0.001 | 0.001 | 0.862 | 0.805 | 0.318 | 0.014 | 0.315 | 0.022 | <0.001 |
| Season | 2 | <0.001 | 0.004 | <0.001 | <0.001 | <0.001 | <0.001 | <0.001 | <0.001 | <0.001 | <0.001 |
| Year | 1 | 0.001 | <0.001 | 0.424 | <0.001 | <0.001 | <0.001 | <0.001 | <0.001 | <0.001 | <0.001 |
| Treatment × season | 6 | 0.288 | 0.243 | 0.012 | 0.37 | 0.368 | 0.003 | 0.005 | 0.352 | 0.007 | 0.007 |
| Treatment × year | 3 | 0.496 | 0.943 | 0.169 | 0.795 | 0.774 | 0.478 | <0.001 | 0.745 | 0.03 | 0.375 |
| Season × year | 2 | 0.827 | 0.011 | 0.134 | <0.001 | <0.001 | <0.001 | <0.001 | 0.002 | <0.001 | <0.001 |
| Treatment × season × year | 6 | 0.039 | 0.05 | 0.596 | 0.017 | 0.331 | <0.001 | 0.02 | 0.594 | 0.009 | 0.302 |

Notes:
For each property, data was pooled from 360 independent samples, e.g., two sampling years × three seasons × four thinning treatments × five soil depths × three repetitions.
Df, degree of freedom; STN, soil total nitrogen; SOC, soil organic carbon; MBN, microbial biomass nitrogen; DON, dissolved organic nitrogen; MBC, microbial biomass carbon; DOC, dissolved organic carbon.

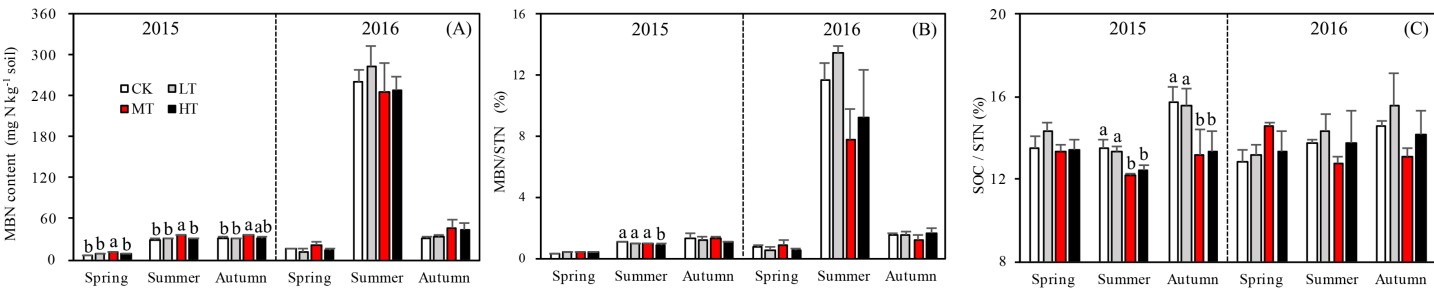

**Figure 4 Variation in the MBN (A), MBN/STN (B), and SOC/STN (C) in different thinning treatments across the growing seasons.** CK, the no thinning control treatments. LT, the low thinning treatments (15% thinning). MT, the moderate thinning sample treatments (35% thinning). HT, the heavy thinning sample treatments (50% thinning). MBN, microbe biomass nitrogen; STN, soil total nitrogen; SOC, soil organic carbon. Each bar represents an average value across three replicate samples ($n = 15$), that is, three plots repeats × five soil depths. Error bars represent standard errors around the three plot repeats. Different lowercase letters demarcate a significant difference among different density adjustments within the same sampling season ($p < 0.05$).

## Soil nitrogen components

Interaction effects among treatment, sampling season and year were found ($p = 0.017$) for DON (Table 4). Seasons and the two sampling years affected DOC more significantly ($p < 0.001$) than treatment with either additional variable. The soil in 2015 accumulated more DON overall than in 2016 and significantly more DON accumulated during summer 2015 than in any other season (Fig. 3A). Tree stem density had little effect on DON in the soil across the sampling seasons. However, DON varied significantly with the seasons ($p < 0.001$), changing rapidly over the sampling period. The DON was 102.7% higher in summer than in the other seasons.

The MBN, which reflects the microbial activity of forest soils, was highest in the MT treatment across all seasons ($p = 0.012$ in spring; $p = 0.076$ in summer; $p = 0.035$ in autumn) in 2015. The mixed-effects model indicated no interaction effects among treatment, sampling season and year ($p = 0.331$) for MBN. However, the interaction between season and year affected MBN significantly ($p < 0.001$; Table 4). The soil in summer 2016 accumulated more MBN than the other five seasons for all forest thinning treatments (Fig. 4A). MBN generally increased with decreasing tree stem density from the

LT to the MT treatment, and then decreased in the HT treatment (mg N kg$^{-1}$):
in spring, CK < HT (7.7 ± 0.79) < LT (8.8 ± 1.16) < MT (10.8 ± 0.30); in summer,
CK (29.9 ± 2.49) < LT (30.5 ± 1.32)) < HT (32.2 ± 2.97) < MT (36.4 ± 0.93); and in autumn,
LT (30.2 ± 0.80) < HT (33.0 ± 0.51) < CK (30.7 ± 3.37) < MT (35.8 ± 0.44). However,
in 2016, MBN was not significantly affected by density adjustment ($p = 0.165$ in spring;
$p = 0.555$ in summer; $p = 0.205$ in autumn). In summer 2016, a significantly higher
MBN was measured, which was 302.6% higher than the average MBN content across
all the seasons and treatments.

### Relationships among soil nitrogen components

The ratios of DON/STN and MBN/STN responded differently to both thinning treatment
and season (Figs. 3B and 4B). No significant interaction effects were found among
treatment, sampling season and year individually ($p = 0.302$) for DON/STN. However,
season and year interacted to affect DON/STN ($p < 0.001$). The soil in summer 2015
accumulated a higher DOC/STN ratio than the other five seasons for all thinning
treatments.

A one-way ANOVA revealed that both ratios varied with season ($p < 0.01$), being higher
in autumn and summer than spring. As noted, the DON/STN ratio varied significantly
with tree stem density in four sampling seasons out of six. DON/STN generally decreased
with decreasing tree stem density from the LT to the MT treatment, and then increased
in the HT treatment (Fig. 3B) (%): spring 2015 ($p = 0.027$), MT (1.42 ± 0.13) < CK
(1.59 ± 0.05) < LT (1.73 ± 0.08) < HT (1.78 ± 0.04); summer 2015 ($p = 0.003$), MT (6.13 ±
0.16) < CK (6.58 ± 0.20) < LT (7.79 ± 0.34) < HT (8.47 ± 0.76); autumn 2015, not
significant ($p = 0.10$); spring 2016 ($p = 0.123$); summer 2015 ($p = 0.047$), MT (2.70 ± 0.43)
< HT (3.22 ± 0.91) < CK (4.18 ± 0.16) < LT 4.51 ± 0.55); autumn 2015 ($p = 0.001$),
MT (2.32 ± 0.15) < HT (3.18 ± 0.22)< CK (3.33 ± 0.23) < LT (3.45 ± 0.13). Each season,
the DON/STN reached its minimum in the MT plot. For MBN/STN, a significant
interaction effect was found among sampling season and year ($p < 0.001$; Table 4).
The data indicated that soils contained the highest MBN/STN in summer 2016 (Fig. 4B).

Strong, positive correlations were found between SOC and STN ($R = 0.894$, $p < 0.001$,
$n = 360$), DOC and DON ($R = 0.926$, $p < 0.001$, $n = 360$), and between the DON and
both the microbial biomass carbon (MBC) ($R = 0.657$, $p < 0.001$, $n = 360$) and DOC
($R = 0.926$, $p < 0.001$, $n = 360$). In contrast, the SOC/STN ratio negatively correlated
with STN ($R = -0.427$, $p < 0.001$, $n = 360$; Fig. 5).

## DISCUSSION

The specific objectives of this study were to determine how STN varies with stand tree
stem density in a *L. principis-rupprechtii* plantation and how variation in each soil nitrogen
component may drive patterns in STN. STN responded to density treatments,
increasing with decreasing density, peaking in the MT treatment (35% tree stem removal),
then decreasing in HT (50% removal), which indicates that thinning generally leads to
increased STN. However, this affected was limited to the growing season and was not seen
in autumn.

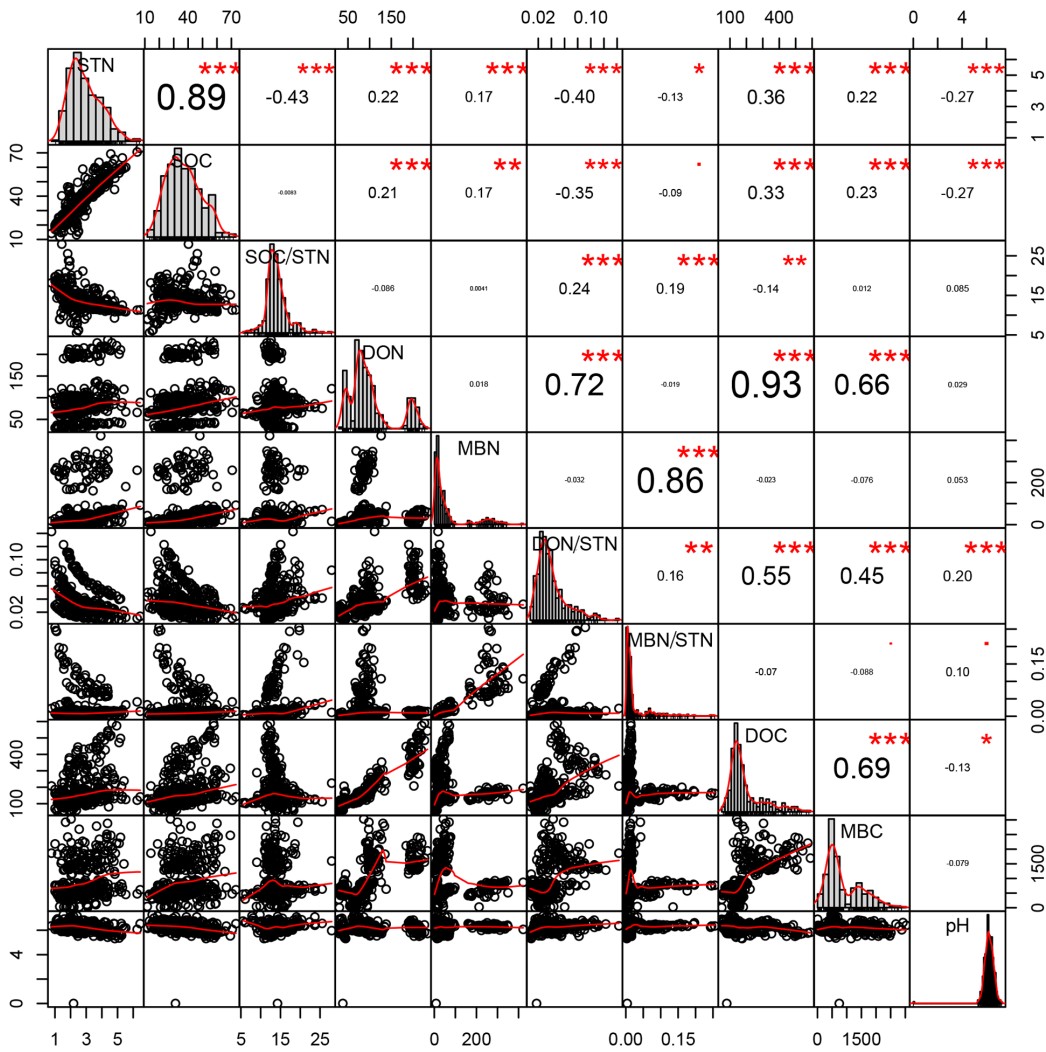

**Figure 5 Pearson relationship of different soil properties across thinning treatments, seasons, and soil depths.** $n = 360$, that is, four density treatments × three seasons × three repeats × five soil depths × 2 years. $^*p < 0.1$; $^{**}p < 0.05$; $^{***}p < 0.01$.

## Environmental factors effect on the soil N pool

The availability of soil N is widely regarded as a factor commonly restricting primary productivity (*Sigurdsson, 2001*) and the function of certain biochemical processes (*Vitousek et al., 2010*). Similarly, an experiment that followed mixed forests for 12 years after thinning showed that tree stem density reduction can significantly improve the growth of woody species in stands (*Lei, 2005*). Studies in *Picea abies* (*Heinrichs & Schmidt, 2009*) and *Pseudotsuga menziesii* forests (*Ares, Neill & Puettmann, 2010*) also found that both forest species richness and the abundance of shrub and grass species increases with thinning intensity. Aboveground vegetation is one of the main sources for the soil N (nitrogen) pool (*Achat, Fortin & Landmann, 2015*), hence changes in species composition and biomass may affect STN. As we found here, understory plant species were most abundant in the MT treatment (Table S1). More understory biomass was found in

plots that had undergone a more intense thinning treatment (Table 3), echoing the higher values of STN and SOC, also in the MT plots.

The forest density adjustment conducted in the *L. principis-rupprechtii* plantation altered the environmental factors both at all layers (light condition, soil respiration, temperature, and moisture) in the soil, which may also contribute to the difference in the soil N pool. Light and space availability in the understory can change with thinning (*Richards & Hart, 2011*; *Roberts, 2004*; Table 1). Here, thinning treatments altered the total PPFD under and had no impact on the total PPFD over (Table 2). Other crucial environmental factors like soil temperature and soil respiration changed with density adjustments. Higher soil temperatures and soil respiration were measured in the MT, though not significant for either sampling year. 2016 was wetter than 2015, with precipitation in 2016 1.3 times higher than 2015 (Fig. 1), which may also explain the significant difference in soil respiration between the 2 years.

The higher soil temperatures in MT throughout the entire growing season suggests that this important environmental factor (*Ma et al., 2010*) may be directly affect by MT. The soil under MT received more radiation, the main energy source, than CK and LT. As for HT, the intensive 50% thinning may have led to the reduction in ground litter, thus scattering the heat to the atmosphere. Soil moisture was enhanced significantly by the treatment, with the increase in thinning leading to a higher soil moisture content (Table 2).

According to the intermediate disturbance hypothesis (*Fox, 1979*; *Roxburgh, Shea & Wilson, 2004*; *Huston, 2014*), moderate rates of disturbance to plant communities can maintain high species diversity. This was observed on an experimental *Cupressus funebris* plantation, where MT enhanced the diversity indices of both understory shrub and herbaceous species (*Gong, Niu & Mu, 2015*). Combinations of various environmental factors, such as understory plant species composition and light and space availability, may alter the soil environment to different extents, thus affecting STN concentrations.

Close relationships were found between STN and other soil properties. Plotting all the data across treatment and season, higher STN concentrations are shown to correspond to higher concentrations of SOC, DON, DOC, and MBN (Fig. 5). *Bravo-Oviedo et al. (2015)* and further analysis performed in this study revealed that density treatments affect various components of the soil N pool, which are considered to be factors driving variation in total soil N.

## Effect of nitrogen components on the soil N pool

This study tested the hypothesis that MT treatments should increase STN through altering (a) the environmental factors in the forests and (b) the soil N-components and the solubility of the N pool.

Changes in DON can lead to significant modifications in soil nutrient stoichiometry, thereby affecting microbial activity and STN concentration (*Iqbal et al., 2010*; *Aerts, Bodegom & Cornelissen, 2012*). Even though there was no significant correlation found between tree stem density and DON, the thinning treatments were found to alter soil nitrogen characteristics, with one unit of STN containing less DON in the more extreme thinning treatments (Fig. 3B). The amount of DON can affect STN dynamics, as a higher

DON/STN means a greater possibility of nitrogen loss through leaching, which would affect nitrogen accumulation rates. The DON/STN ratio was affected significantly by thinning treatment ($p < 0.001$, Table 3) and was at its minimum in MT plots (Fig. 3B), the same treatment where the highest concentrations of STN were recorded in spring and summer 2015, summer and autumn 2016 (Fig. 2A). This may partly explain the variation in STN across treatment type, with higher STN values where the DON/STN ratio was lower.

Here, moderate tree stem density reduced nitrogen solubility, limiting nitrogen losses. An analysis of hydrological characteristics in the study area revealed abundant rainfall, which may have caused fertilizer to wash away (Figs. 1 and 3). The effects of the MT treatment on the DON/STN ratio matched expectations, however, DON alone did not.

Total soil DON and the DON/STN ratio varied with season (Fig. 3). In summer, as the temperature gradually increased (Fig. 1), trees and grasses would have experienced abundant root growth, likely leading to an increase in root secretions and the amount of deciduous material surrounding the root system (*He, Zhou & Li, 2013*). Both soil STN and DON concentrations rose from spring to summer. Soil temperature and precipitation were also greater in the summer (Fig. 1). Microbial growth can expand in the presence of increased temperatures (*Edwards et al., 2006*), which can then be further facilitated by higher concentrations of DON, providing more nitrogen for the microbes (*Iqbal et al., 2010*). In this study a close positive relationship was measured between DON and MBC ($R = 0.657$, $p < 0.001$, $n = 360$; Fig. 5). Increased precipitation has been reported to be an important factor affecting the N pool (*Yu et al., 2017*) and the resulting nitrogen (DON) losses from summer to autumn might explain the STN reduction in autumn. This confirms the hypothesis that MT reduces the DON/STN ratio, thus, enhancing the STN (Figs. 2A and 3B).

Tree stem density had a more complex effect on microbial indices like MBN, MBC, and SOC/STN. MBN was 302% higher in summer 2016 than the MBN averaged over the 2 years. The MBN concentration tended to increase with decreasing tree stem density, reaching its highest level in the MT treatment before decreasing in HT. This pattern was echoed by the STN concentration (Figs. 2A and 4B), though was only significant in spring and summer 2015. In 2016, when there was 130% more precipitation, MBN was not affected by thinning treatment.

The MBN concentration and MBN/STN ratio were much higher in summer and autumn than in spring (Fig. 4), as has also been found in previous studies on temperate forests (*Bohlen et al., 2008*), which indicates that microbial activity is lower at the start of the vegetative season. Adequate water availability and warmer temperatures for likely augmented microbial growth and produced the observed increase in MBN during the summer (Figs. 1 and 4A). The observed average MBN/STN ratio (2.5%) was similar to the other temperate forest soils (1–3%) (*Zhong & Makeschin, 2006*).

A previous study has indicated that a lower SOC/STN ratio indicates an increment of the rate of microbial decomposition and of nitrogen mineralization (*Springob & Kirchmann, 2003*). Here, the SOC/STN ratio was negatively correlated with STN (Table 4). The MT treatment likely provided a better environment for microorganism growth, thus enhancing the rate of microbial decomposition. Greater microbial biomass could then

increase the concentration of MBN because MBN and MBC were strongly positively correlated, as shown in the Pearson relationships (Fig. 5). Soil microbial biomass (as MBC or MBN) can be sensitive to either changing soil conditions, a slight variation in the composition of soil organic matter (Liu, 2010) or environmental (Yi et al., 2007) changes. Hence, MBC and MBN have been suggested as an index of both soil environmental change and nutrient supply capacity (Hargreaves et al., 2003). The highest MBN (Fig. 4A) and the lowest SOC/STN (Fig. 4C) were observed in the MT treatment, suggesting that microbes might have been more active under intermediate tree stem densities.

## CONCLUSIONS

Clear effects of thinning treatments were found on STN in a *L. principis-rupprechtii* plantation 3 years after thinning. The STN concentration was greatest in the MT plots. MT treatments may have enhanced the accumulation of the soil N pool by changing (a) the environmental factors and (b) the solubility of the soil N pool. The influence of the thinning treatments on the N pool are likely driven by the effect of stem density on the labile N pool. For example, DON and MBN varied with season and peaked in the summer when the water and heat conditions were better for the carbon cycle. A lower DON/STN ratio was found under intense thinning in response to a higher STN content, which indicates that the solubility of the soil N pool was changed by the density treatments. The lower solubility spurred by the MT treatment is the key factor causing greater STN accumulation. The environmental factors of soil temperature, soil moisture and light moderated conditions to be better for microorganisms and the plants, which also contribute to the STN accumulation. We recommend MT (1,404 trees per ha) of *L. principis-rupprechtii* plantations to promote N retention and we agree with the *intermediate disturbance hypothesis*, but suggest that long-term studies are still required to validate these findings.

## ACKNOWLEDGEMENTS

We gratefully acknowledge support from the Taiyue Forestry Bureau and the Haodifang Forestry Centre for fieldwork. We also thank all those who provided helpful suggestions and comments on improving the quality of this manuscript. We would also like to thank Elizabeth Tokarz at the Yale University for her assistance with English language and grammatical editing of the manuscript.

### Funding

This study was supported by the National Key Study and Development Program of China (NO. 2016YFD0600205). The funders had no role in study design, data collection and analysis, decision to publish, or preparation of the manuscript.

### Grant Disclosures

The following grant information was disclosed by the authors:
National Key Study and Development Program of China: 2016YFD0600205.

## Competing Interests

The authors declare that they have no competing interests.

## Author Contributions

- Junyong Ma conceived and designed the experiments, performed the experiments, analyzed the data, prepared figures and/or tables, authored or reviewed drafts of the paper, approved the final draft.
- Hairong Han contributed reagents/materials/analysis tools, authored or reviewed drafts of the paper, approved the final draft.
- Wenwen Zhang performed the experiments.
- Xiaoqin Cheng analyzed the data, authored or reviewed drafts of the paper.

## Data Availability

The raw data are provided in the Supplemental File.

## Supplemental Information

Supplemental information for this article can be found online at http://dx.doi.org/10.7717/peerj.5647#supplemental-information.

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
