# Peer review of "Dynamics of nitrogen and active nitrogen components across seasons under varying stand densities in a Larix principis-rupprechtii (Pinaceae) plantation"

_PeerJ, doi:10.7717/peerj.5647_

## Round 0.1 · original submission · Minor Revisions

Both authors have provided detailed and constructive reviews. Please take serious consideration of these reviews in the revision. I will also add that the stats in Table 4 is not correct: time (year, season) should be repeated measure, and depth is nested with the thinning treatment. A simple four-way ANOVA is not valid for this kind of analysis. Mixed-effects model should be used to consider the complicated relationships among these factors (year, season, depth) and treatment.

Also, the language and writing should be much improved (as mentioned by both reviewers). The current manuscript has many errors which need to be fixed.

Reviewer 1 ·

Basic reporting

Basically, it is easy to read and understand. But I have hard time to know exactly the meaning of some sentences. “Dissolved organic nitrogen … nutrient availability” (at Line 53-56) might be problematic in grammar. Line 227-229 is confusing. Please ask the English native writer to help modify the whole paper if possible.
I am not sure if I understand your third question right (at Line 83). Do you mean how the environmental factors influence the N pools?
Some typo errors: Line 19, 146, 152, 209, 235, 249-250.
It is very vague to say “more and less severe thinning treatments”? (Line 216) Could you please provide a clearer way?
Since the authors want to compare the N components at two aspects - treatments and seasons, I would suggest the authors to illustrate the comparisons in order, that is, effects of treatments first and effects of seasons second; or vise verse. It might be clearer and easier for reader to follow and compare.
For a clear frame of paper, I might suggest the authors to list sub-titles in Discussion which could correspond the sub-titles in Result.
At Line 358, you mentioned “enhance the soil N pool”, did you mean to enhance the accumulation or transformation of soil N pool. It is better if you give a clear point here.
I was confused by your labels on x-axis in the figure 1, so please double check the labels.

Experimental design

Could you provide the descriptive characteristics of the background information (e.g. altitude, grade, slope direction and soil condition, etc.) of the stands you chosen for quadrats?
What is the scientific or empirical criterion to set the ratios of thinning?
I am confused by your quadrat design. As I think you have 3 quadrate which includes 12 study treatments, but you said “three sample area (which equals to quadrat) was designed for the 15% thinning (Line 103-104). Could you please explain it? Or could you please provide plot design graphs to illustrate?
For the measures of light environment, could you please explain why you choose July? What was the time in a day as you measured? And how many times did you conduct the measures in 2015 and 2016, respectively?
As many factors, besides the main interest of thinning, would impact the soil N and C cycles, so I might suggest the authors to consider some other statistical methods to identify the importance among those factors, like redundancy analysis (RDA).

Validity of the findings

Why the soil temperature was higher in MT in 2015? Do you have a reasonable explanation? (Line 197)
How did you calculate the total biomass at Line 197?
For the understory, did the local clear the weeds or understory? If yes, do they use herbicide or other ways?
Did you apply fertilizer to the plantation? If yes, what is the amount and kind of fertilizer?
What is your purpose to add figure 6? What are the potential impacts of runoff and sediment concentration on your current results?
In table 2, it looks like the soil respiration shows obvious difference between 2015 and 2016. Could you use paired t-test to examine if these environmental factors statistically differ between 2015 and 2016?
I would suggest you enhance your results interpretation and discussion based on table 4. Because table 4 is actually the core to support your research and findings.

·

Basic reporting

The authors reported on how the process of thinning in forests could positively or negatively impact the soil nitrogen and carbon profiles. The literature cited within this research project is sufficient for the objectives and clearly states the requirement of thinning within agricultural forestry practices. The research rational is very applicable to the forestry industry and the hypotheses regarding thinning and soil interactions have been established. The structure and visual elements of the manuscript is acceptable. Authors should consider their figure legends to be of international reporting standard (figures labeled at the bottom, tables at the top). Some figures are of poor quality which hinders the readers ability to clearly distinguish. Within the review manuscript, tables are incomplete (it is cut off toward the end of the page). Figures should be re-submitted in higher quality format.

However, there are some grammatical errors within the document and it would be advised to send the manuscript for English grammar correction. Authors should also note that there are clear differences in font size throughout the manuscript and line spacing is inconsistent, which needs to be addressed. Authors should also consider placing their equations within a text box, or at least in the centre of the page.

Experimental design

The primary research objectives fits within the scope of PeerJ and the authors stated their research rational. This rational is relevant within the agricultural forestry industry and could potentially have a positive impact on future ecological and conservation ventures. The process of thinning (and how the authors compared the various degrees of thinning) and its impact on soil nitrogen and carbon flux provides a meaningful contribution to the knowledge within the specified field. The experimental design addresses all the necessary components (including microbial) which could influence soil nitrogen and carbon content and the experimental process has been given in enough detail to reproduce.

Validity of the findings

This study looked toward potential strategies for the improvement of soil nitrogen and carbon assimilation, by looking at thinning processes within plantations. It was concluded that moderate thinning could potentially be of great value, when applied, to species diversity and assisting in the growth of a variety of organisms above- and under-ground. This interaction between species is very important in maintaining the soil nitrogen and carbon content.

The authors clearly stated their conclusion and this study could hold very meaningful outcomes for future forestry practices. Findings are valid and, the authors acknowledge the need for further research within their hypotheses.

Additional comments

No general comments. I wish for the above-mentioned review comments to be sent to the authors.

---

## Round 0.2 · accepted · Accept

I am convinced the revisions significantly improved the manuscript and so I am happy to accept your manuscript for publication. Just a small detail, I think the caption of the table 4 was not updated (still mentioning ANOVA).

There are some additional minor typos identified by Reviewer 3, which you should address while in Production.

# Reviewer 3 ·

Basic reporting

no comment

Experimental design

no comment

Validity of the findings

no comment

Additional comments

General comments: I have read the manuscript “Dynamics of nitrogen and active nitrogen components across seasons under varying stand densities in a Larix principisrupprechtii (Pinaceae) plantation” (Ma et al). Undoubtedly, the present manuscript has been improved significantly after careful revision. Authors conducted measurements of soil nitrogen and its active components during the growing seasons and corresponding environmental variables with four treatments, and performed appropriate statistical analyses. This study indicates effects of forest management (thinning) on soil nitrogen pool in a larch plantation and the moderate thinning exhibits more important role for the accumulation of soil nitrogen pool by adjusting a series of factors. The manuscript is well organized, the results are interesting and fits to the scope of the Journal.

Specific comments:
Line 19: soil respiration?
Lines 27-28: I think these conclusions should be rewritten, because the conclusion is based on your findings rather than inference
Line 135: Why did you conduct only one day’s measurement of soil respiration?
Line 268: affected
Line 294: and
Line 307: There should be a blank before and after “=”, just like before, and this should be consistent throughout the manuscript
Line 328: delete the comma
Line 343: unclear, is that mean the environmental factors in moderated conditions would be better for microbes and plants?